# Potential Human Exposure to Mercury (Hg) in a Chlor-Alkali Plant Impacted Zone: Risk Characterization Using Updated Site Assessment Data

**Symbat Kismelyeva** [1], **Rustem Khalikhan** [2,†], **Aisulu Torezhan** [2,†], **Aiganym Kumisbek** [3], **Zhanel Akimzhanova** [4], **Ferhat Karaca** [2] 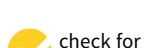 **and Mert Guney** [2,*]

1. Environmental Science and Technology, Faculteit Bio-Ingenieurswetenschappen, Universiteit Gent, Coupure Links 653, 9000 Gent, Belgium; symbat.kismelyeva@ugent.be
2. Department of Civil and Environmental Engineering, School of Engineering and Digital Sciences, The Environment & Resource Efficiency Cluster (EREC), Nazarbayev University, Kabanbay Batyr Avenue 53, Nur-Sultan 010000, Kazakhstan; rustem.khalikhan@nu.edu.kz (R.K.); aisulu.torezhan@nu.edu.kz (A.T.); ferhat.karaca@nu.edu.kz (F.K.)
3. Faculty of Engineering, University of Debrecen, Otemeto Street 2-4, 4028 Debrecen, Hungary; aiganymkumisbek@gmail.com
4. Environmental Engineering and Sustainability Management, École Polytechnique de Paris, Route de Saclay, CEDEX, 91128 Palaiseau, France; zhanel.akimzhanova@polytechnique.edu
* Correspondence: mert.guney@nu.edu.kz
† These authors have equally contributed to this work.

**Abstract:** Industrial activities have resulted in severe environmental contamination that may expose rural and urban populations to unacceptable health risks. For example, chlor-alkali plants (CAPs) have historically contributed mercury (Hg) contamination in different environmental compartments. One such site (a burden from the Soviet Union) is located in an industrial complex in Pavlodar, Kazakhstan. Earlier studies showed the CAP operating in the second half of the twentieth century caused elevated Hg levels in soil, water, air, and biota. However, follow-up studies with thorough risk characterization are missing. The present study aims to provide a detailed risk characterization based on the data from a recent site assessment around the former CAP. rThe $\Sigma HI$ (hazard index) ranged from $9.30 \times 10^{-4}$ to 0.125 (deterministic method) and from $5.19 \times 10^{-4}$ to $2.54 \times 10^{-2}$ (probabilistic method). The results indicate acceptable excess human health risks from exposure to Hg contamination in the region, i.e., exposure to other Hg sources not considered. Air inhalation and soil ingestion pathways contributed to the highest $\Sigma HI$ values (up to 99.9% and 92.0%, respectively). The residential exposure scenario (among four) presented the greatest human health risks, with $\Sigma HI$ values ranging from $1.23 \times 10^{-2}$ to 0.125. Although the local urban and rural population is exposed to acceptable risks coming from exposure to Hg-contaminated environmental media, an assessment of contamination directly on the former CAP site on the industrial complex could not be performed due to access prohibition. Furthermore, the risks from ingesting contaminated fish were not covered as methyl-Hg was not targeted. An additional assessment may be needed for the scenarios of exposure of workers on the industrial complex and of the local population consuming fish from contaminated Lake Balkyldak. Studies on the fate and transport of Hg in the contaminated ecosystem are also recommended considering Hg methylation and subsequent bioaccumulation in the food chain.

**Keywords:** exposure assessment; heavy metals; Kazakhstan; Pavlodar; risk assessment; sediment contamination; site characterization; soil pollution; water pollution

## 1. Introduction

Consequences of rapid industrialization with far-reaching impacts include discharges of potentially toxic elements (PTEs) into the environment and their subsequent accumulation in various environmental sinks [1]. Mercury (Hg), a ubiquitous element, is a PTE

classified as a priority hazardous substance according to the adopted Water Framework Directive and the Environmental Quality Standards Directive of the European Commission [2–4]. Hg and its compounds are considered persistent and bioaccumulative [5]. Hg exists in the environment in three main forms: elemental ($Hg^0$), organic (e.g., methyl Hg (Me-Hg)), and inorganic (e.g., $Hg^{2+}$) [6,7]. Despite the ability of Hg to be transformed into its other more toxic forms, elemental Hg cannot be broken down or reduced into less harmful substances [6]. Inorganic Hg can also enter water or soil through weathering of Hg-bearing rocks, Hg-containing discharges from industries or water treatment facilities, and combustion of municipal waste with considerable Hg content (e.g., thermometers, electrical switches, batteries). Me-Hg is the most common Hg-containing organic compound with high toxicity and bioavailability [8]. In the environment, organic forms of Hg are found in extremely small amounts, between 0.006 and 0.02 $\mu g/m^3$ in outdoor air [8]. The other forms of Hg can enter and accumulate in the food chain via microorganisms (bacteria, phytoplankton, and fungi) and natural processes that can generate Me-Hg from these forms in certain edible fish and marine mammals [8].

Hg and its common compounds (i.e., HgS (cinnabar), $HgCl_2$) can enter the environment through a variety of natural mechanisms in the Earth's crust. Anthropogenic activities such as agriculture and overland traffic, in addition to geogenic sources of PTEs, including Hg, can increase the severity of site contamination [9–11]. Anthropogenic sources of Hg include combustion of fossil fuels, mining activities, electricity-generating power stations, incineration, and production of chlorine along with caustic soda by chlor-alkali plants (CAPs) based on the Hg-cell technology. In contrast, natural sources include volcanic activities and biological and weathering processes [7]. Natural levels of Hg deposited in the environment are relatively low, with the exception of Hg-containing ores. In contrast, the quantity of Hg detected in soil and water in the proximity to hazardous waste sites from anthropogenic activities might be considerable (up to >200,000 times of natural levels) and account for one to two-thirds of the total Hg released into the environment [7]. Released Hg, mostly in the form of metallic and inorganic Hg, may be transported over great distances and could remain in the atmosphere after being released, turning localized discharges into global concerns. Atmospheric Hg can be further converted into different forms and deposited to water and soil via rain or snow. Surface soils, water bodies, and bottom sediments act as sinks for directly released elemental Hg from industrial activities [7], significantly impacting local regions [12,13].

Human exposure to Hg is possible via the following pathways: ingestion of contaminated media, inhalation of Hg vapor or contaminated dust, and dermal contact with contaminated media [14]. Hg exposure causes severe adverse health issues depending on its magnitude, dose, and duration [8]. Children form a particular sub-group due to age-specific behavior to play with soil, the possible presence of pica behavior (ingestion of non-food items), and toxicodynamic and toxicokinetic differences such as higher body surface area to volume ratio and larger consumption of water per unit of body weight [5,14,15]. Children's exposure to Hg may result in developmental impairment, including intellectual deficiency, speech disorders, and sensory issues, whereas adult exposure causes various nervous system damages [16,17]. The "minimal risk level" for chronic Hg inhalation is stated as 0.2 $\mu g/m^3$, and for oral ingestion of inorganic Hg as 0.0003 mg/kg/day [8]; the "reference dose" (*RfD*) for oral ingestion of inorganic Hg is suggested as 0.001 mg/kg/day by the USEPA [5].

CAPs based on Hg-cell technology were considered one of the world's major concerns, contributing to emissions of around 4378 kg Hg/y [18] until 2017, the year restrictions based on the Minamata Convention entered into force [6]. Even though chlor-alkali sector is moving towards energy-efficient, Hg-free membrane technologies, a complete immediate shutdown of Hg-emitting operations is not conceivable [6]. In Kazakhstan, Hg was used as a cathode in an electrolytic separation process of the product from brine at the former "PO Khimprom" CAP (former Soviet Union military industrial establishment), producing large quantities of Hg-rich sludge. Chlorine and caustic soda manufacturing for both

civilian and military purposes was carried out employing 80 electrolytic cells, each of which contained about 2700 kg of Hg [19,20]. In the period of 1975–1993, the former CAP in the Pavlodar area released roughly 135 t of Hg into Lake Balkyldak, and undetected Hg losses totaled approximately 1000 t [21,22]. Previous studies have reported elevated concentrations of Hg in the area impacted by the former CAP in the Pavlodar region and attempted to assess associated health risks (e.g., [23,24]). A complete list of these studies along with their detailed discussion was presented in our recent review article on this contaminated site [9]. As no recent study provided a much-needed update on the environmental situation in Pavlodar, Kazakhstan, we have addressed this gap and recently published our findings [25,26]. Furthermore, a detailed comprehensive risk assessment on the current situation in the region is still not present. It is important to estimate human exposure to Hg from contaminated environmental compartments, as our recent site investigation indicated extensive contamination in different environmental media at certain locations [25,26]. Therefore, the present study aims to characterize human health risks from potential exposure to Hg in contaminated air, water, sediments, and soil, considering possible exposure scenarios for children and adults.

## 2. Materials and Methods

### 2.1. Study Area, Sampling

The site assessment was performed in the Pavlodar region, which includes a large industrial complex that is a part of the "Pavlodar-Aksu-Ekibastuz" territorial industrial complex of Kazakhstan. As of 2019, the number of industrial enterprises in Pavlodar amounted to about 900 [27]. The Hg pollution near Pavlodar city (52°18′56″ N, 76°57′23″ E) is the result of a former CAP "Khimprom" operated from 1975 to 1993 [19]. Lake Balkyldak, located to the north of the former CAP, is considered as one of the principal pollution sources due to serving as a settling lagoon in the past, where the majority of the Hg-containing sludge has been discharged [19]. The Pavlodar region has an extreme continental climate characterized by dramatic seasonal variations and high evaporation rates [22]. The soils around the study region are classified as kastanozems [28], which may be expected to retard contamination propagation. A total of 129 solid (soils, sediments) and 98 aqueous (surface water, groundwater) samples were collected along with 143 ambient air Hg measurements performed during multiple field trips to the region in 2018 and 2019. Samples were collected in May (2018), September–October (2018), and July (2019). During soil and sediment sampling, a composite sample from 0–10 cm depth was collected. Aqueous samples were collected in 15 mL and 50 mL tubes. All field samples were transferred to the laboratory and then stored at −4 °C.

### 2.2. Physio-Chemical Characterization of Solid and Aqueous Samples

Solid samples were characterized for their pH level, among which eight randomly selected soil samples were characterized for their total carbon (TC), total organic carbon (TOC), and total nitrogen (TN). Soil pH was measured according to ASTM Method D 4972-95a (1995) [29]. TC and TOC was measured using a C/N dry combustion elemental analyzer (Multi analyzer HT 1300; Analytik Jena, Jena, Germany). TN was determined according to the Dumas method (using Dumas analyzer DuMaster D-480; Buchi, Switzerland). A total of 57 selected aqueous samples have been characterized for their pH and 23 for conductivity. Conductivity was measured in situ, and pH was measured in the laboratory according to ASTM Method D 1293-99 (2005) [30].

### 2.3. Determination of Hg Content

The measurements of total Hg content were performed using the RA-915M Hg analyzer coupled with RP-92 and PYRO-915+ attachments (Lumex, IL, USA) with a low detection limit (i.e., 0.5 ng/L for aqueous samples, 0.5 µg/kg for solid samples, 0.5 ng/m$^3$ for ambient air measurements) [31]. The PYRO-915+ attachment was used to measure total Hg in solid samples, which were tested in duplicates or triplicates if the difference between

recorded Hg levels exceeded 15%. The cold vapor technique based on bromide–bromate mineralization was used to determine total Hg in the aqueous samples with the RP-92 attachment [31]. For direct air Hg measurements, the portable Hg analyzer (RA-915 Portable Zeeman Hg Analyzer) was used in situ. The quality control procedure involved both the internal calibration check performed prior to each field trip and periodic zero check sets at every 10-minute interval.

### 2.4. Exposure Assessment and Risk Characterization

Exposure to Hg was assessed considering four scenarios: recreational activities, fishing, residential exposure, and soil contact-intensive work. These scenarios included one or more possible oral, inhalation, and dermal contact pathways. The ingestion of soil and sediment particles has been considered as the most probable pathway due to the availability of soil particles to incidental ingestion caused by hand-to-mouth contact behavior [11,32]. Since humans can be exposed to Hg in water in a variety of scenarios, exposure assessment in the present study included surface water ingestion and dermal contact pathways. Air inhalation was also a pathway of interest because people were exposed to volatile Hg in all scenarios. Risk characterization included separate scenarios for both CAP-affected and urban areas of the region. Recreational activities (i.e., swimming and playing in water) and fishing were identified as the most critical and probable scenarios for the CAP-affected zone. Residential exposure and soil contact-intensive work were selected as the most critical and probable scenarios for urban areas.

Deterministic and probabilistic approaches were employed for conducting a non-carcinogenic human health risk assessment according to the most recent data from USEPA manuals [33–35]. The formulas used for calculating these parameters as well as the parameters selected used for deterministic risk assessment are presented in Table A1 (see Appendix A). The pathway-specific values of these parameters are presented in the Supplementary Materials (Tables S1–S4). Exposure duration (ED) value for the scenario of residential exposure was selected as 350 days, as recommended by USEPA for characterizing risks for residential receptors. The averaging time (AT) was calculated for one year of exposure ($365 \times 24 \times$ ED). Similarly, the averaging time for sub-chronic exposure pathways was determined for a complete week of exposure ($24 \times 7 \times$ ED). Bodyweight (BW) values used in the calculations were taken per USEPA recommendations [36] as these were the best available data, which may be slightly higher than the average BW values of different age and gender groups of the population of the Republic of Kazakhstan [20]. The ingestion rate (IR), skin surface area available for contact (SA), and contact rate (CR) values were obtained from the USEPA Exposure Factors Handbook [37]. For the recreational scenario, sediment IR values for exposure during swimming were used. These are more representative of the oral Hg exposure as it was shown that people ingest more soil during recreational activities in water bodies due to soil and sediment resuspension [37].

A thorough parameter selection was performed for both deterministic and probabilistic approaches. A wide range of input parameters could be impacted by uncertainty and variability, making the probabilistic approach useful for exposure models within the risk assessment model [38–40]. The uncertainty is related to the lack of information regarding the actual value of a parameter, whereas variability is related to the intrinsic variation of the data value [41]. In contrast to the deterministic approach, the probabilistic risk assessment is based on probability density functions (PDFs), which describe the input data and model parameters [42]. Probability functions consider uncertainties of input data and model parameters, which propagate throughout the evaluation using probabilistic analysis [42]. It results in the probability distributions for the daily intake values and subsequent individual risk in terms of hazard indices for adults and children. The @RISK 5.5 software [43] was used to perform probabilistic simulations along with introducing the PDFs to probabilistic human health risk assessment. @RISK 5.5 performs probabilistic simulations by varying input values to the best-fitting distributions of each parameter via the Akaike Information Criterion, then running the calculations in the risk assessment

model 10,000 times and producing a probability distribution of output values [44]. The calculations of daily intake (chemical daily intake (*CDI*), exposure concentration (*EC*), and dermally absorbed dose (*DAD*) for ingestion, inhalation, and dermal contact, respectively), *HI*, and Σ*HI* for scenarios were performed on the @RISK 5.5 software, as well.

In the present paper, the values of Hg concentrations were best fitted to several probability distributions (including lognormal, normal, and triangular) using the NCSS statistical software. The Hg concentrations in air, soil, and water were matched to a lognormal distribution function using @RISK. For selected parameters including soil ingestion rate (IR), exposure frequency (EF), exposure time (ET), BW, skin surface area (SA), and contact rate (CR), separate PDFs were used in probabilistic simulations. For the scenario of soil ingestion, IR and BW were set as lognormal distribution functions, while EF was set as a Pert distribution. The primary rationale for characterizing soil IR as a lognormal PDF was the analogous example for the probabilistic activity modeling approach by the USEPA [37]. Similarly, EF was set as a Pert distribution function and ET as a triangular PDF for the air inhalation scenario. For the dermal contact and surface water ingestion pathways, SA and CR were characterized as lognormal PDF. The selection of lognormal distribution for CR was justified by the same example of the USEPA as for IR, while lognormal PDF for SA was selected because of the lognormal pattern of the SA curve constructed from the data on skin surface provided by the USEPA [45].

With the conservative assumption of the estimated overall intake being equivalent to the actual dose, human health risk was then characterized using the hazard index *HI*, which was calculated by dividing the *CDI*, *EC*, and *DAD* values for each pathway by *RfD* or reference concentration (*RfC*) [33]:

$$HI_{ingestion} = \frac{CDI}{RfD} \tag{1}$$

$$HI_{inhalation} = \frac{EC}{RfC} \tag{2}$$

$$HI_{dermal\ contact} = \frac{DAD}{RfD} \tag{3}$$

For *RfD* of Hg, 0.0003 mg/kg/d was utilized [5]. An *RfC* of 0.0003 mg/m$^3$ was used for inhalation as given in the IRIS database [17]. For each scenario, the sum of *HI* values (Σ*HI*) for all different exposure pathways involved was calculated and presented as total non-carcinogenic risk. Σ*HI* ≥ 1 was taken as indicating a risk of negative health effects from the exposure to contaminated media [33–35].

## 3. Results and Discussion

### 3.1. Soil Characterization: Hg in Soil, Sediments, Water, and Air

Hg concentrations of the environmental samples with their physicochemical characteristics have been previously analyzed by our research group, and the results are reported in detail in our recent publications [25,26]. In summary, the pH level of the selected solid samples ranged from 7.11 to 9.25, and the majority were categorized as slightly alkaline (38 out of 59), indicating low migration potential of the inorganic elements within the media [46]. The presence of organic matter in the soil was characterized by the contents of TC (2.21–12.82 g/kg), TOC (2.14–10.79 g/kg), and TN (0.033–0.167%) in the selected samples. The carbon-to-total nitrogen ratio was 8.96, which is below 30, associated with a low metal accumulation [47].

The average pH level of the selected aqueous samples was 8.27, where most of the samples (*n* = 41) had a pH level higher than 8, indicating lower solubility of the contaminants [48,49]. In addition, conductivity was measured in 23 randomly selected aqueous samples to characterize the quantity of the dissolved substances. The results ranged from 1.32–43.4 mS/cm with an average conductivity of 9.57 ± 9.13 mS/cm being higher than a typical value for freshwater surface streams <1500 µS/cm) [50].

A summary of Hg concentrations in the analyzed soils, sediments, surface waters, groundwater, and the air (Table A2) showed that the total Hg in the soil samples (*n* = 111) ranged from 0.006 to 6.98 mg/kg. The mean value (0.318 mg/kg) was below the national maximum permissible concentration (MPC) of 2.1 mg/kg. Four samples from the north of Lake Balkyldak exceeded the defined limit. Total Hg concentrations in the sediments (*n* = 18) ranged from 0.004 mg/kg to 24.07 mg/kg, with the mean (2.081 mg/kg) around the national MPC for soil. Three sediment samples collected from Lake Balkyldak had elevated concentrations (4.28–24.1 mg/kg), supporting the conclusion from previous studies that the vicinity of the former CAP, including the Lake Balkyldak, is a potential Hg source [9,22,24].

Overall, Hg concentrations in 44 surface water samples ranged from 0.003 to 724 μg/L, with a mean value of 26.4 μg/L. Seven samples exceeded the 0.5 μg/L threshold (the national MPC for Hg in drinking water) up to 50 times. As for groundwater samples (*n* = 69), even though the majority of the samples complied with the national MPC, 15 samples collected in the vicinity of Lake Balkyldak and Sarymsak Pond had extremely high concentrations that exceeded the MPC up to 186 times (mean: 93.3 μg/L, range: 0.004 to 1340 μg/L).

The ambient air measurements showed elevated THg concentrations in Pavlodar city as well as around Balkyldak Lake [25]. The concentrations varied from 1 to 37 ng/m$^3$ with a calculated urban background concentration of 4.9 ng/m$^3$. The values from the area around the former CAP (16–22 ng/m$^3$) were higher than that in the urban area (4–10 ng/m$^3$) [25]. These elevated concentrations might indicate the presence of significant hazards for the population of the Pavlodar area and the aquatic system nearby.

### 3.2. Human Health Risk Characterization via the Deterministic Method

Tables S5 and S6 provide the estimated values of *CDI*, *EC*, and *DAD* for all four scenarios for both population subgroups, adults and children, for ingestion, inhalation, and dermal contact routes. The output values of exposure assessment and risk characterization, as shown in Tables A3 and A4, indicate that estimated daily intake values for Hg ranged from an order of $10^{-9}$ to $10^{-6}$ mg/kg/d for *CDI* and *DAD*, and from an order of $10^{-4}$ to $10^{-2}$ μg/m$^3$ for *EC*. Σ*HI* values were below 1: between $9.30 \times 10^{-4}$ and $1.28 \times 10^{-2}$ for the 50th percentile, $1.67 \times 10^{-3}$ and $1.76 \times 10^{-2}$ for the 75th percentile, $4.13 \times 10^{-3}$ and $3.81 \times 10^{-2}$ for the 95th percentile, and $5.35 \times 10^{-3}$ and 0.125 for maximum concentrations in each environmental compartment. The obtained results indicate that exposure to elevated Hg concentrations in urban and CAP-affected zones of the Pavlodar region poses a non-carcinogenic risk within the acceptable limit under normal, conservative, and maximum reasonable exposure cases. Although Σ*HI* values are below 1, the risk levels reached 0.125 for children in the residential exposure scenario (i.e., an intake of 12.5% of the *RfD*). People, especially children who are particularly vulnerable to Hg exposure due to their relatively lower body weight and deliberate hand-to-mouth behavior, could already be exposed to Hg from several other sources apart from the scenarios considered in the study [8,14]. As expected, the risk values were consistently higher for children than for adults in all cases.

Referring to Table A5, for the scenarios in the urban zone, namely, residential exposure and soil contact-intensive work, the inhalation pathway was the most significant contributor to the risk values, generating 13.4–99.9% of the overall *HI*. Soil and dust ingestion had a contribution of approximately 0.06–1.63% and 0.23–6.25% for normal and conservative exposure, respectively. This slightly increased up to 3.85–26.5% and 1.27–12.3% for maximum reasonable and maximum concentration exposure. The water ingestion and dermal contact routes were excluded from the risk assessment in urban-zone scenarios because exposure to contaminated water sources (i.e., Lake Balkyldak, Sarymsak Pond, and small ponds nearby) is negligible for citizens of Pavlodar city. This is explained by the lack of the basic infrastructure connecting these water bodies and Pavlodar.

In the cases of recreational activities and fishing, the inhalation route still contributed to risk to a great extent, from 74.4% to 96.0%, followed by soil ingestion, up to 1.60–20.3% for normal exposure. The soil and dust ingestion route added considerably to the risk

with 7.01–53.5% and 20.8–78.5% of the total *HI* under conservative and maximum reasonable exposure scenarios, respectively. It should be mentioned that the pathway of soil and dust ingestion generates about seven times higher risk values for children (<26.5%) than for adults (<3.85%) under more conservative scenarios. The notable difference in this contribution could be attributed to deliberate hand-to-mouth behavior prevalent among children [11,14,51], resulting in higher soil ingestion rates. The water ingestion and water dermal contact pathways both contributed to Σ*HI* by the lowest margins, varying from 0.39% to 4.11% and from 0.73% to 2.76%, respectively. Such contributions from these pathways may be associated with the limited bioavailability of inorganic Hg in Balkyldak Lake. Nevertheless, the water ingestion pathway with a contribution of 4.11% may still be a matter of concern for children in maximum reasonable exposure cases because they are more likely to ingest larger volumes of water during recreational activities (i.e., swimming, water play, diving). It should be noted that risks from exposure to organic Hg bioaccumulated and biomagnified in biota are not considered, which may result in underestimation of the overall *HI* values.

Given that all deterministic parameters utilized mean or average values that could not have been applied to all scenarios, it is possible that the results from deterministic risk assessment overestimated the health hazards related to Hg contamination in the Pavlodar region. To avoid this, it is suggested to rely on the probabilistic risk assessment, which utilizes probability density functions (PDFs).

### 3.3. Human Health Risk Characterization via the Probabilistic Method

Tables A6 and A7 (along with Tables S7 and S8) present the results of probabilistic human health risk characterization for ingestion, inhalation, and dermal pathways for all four scenarios for adults and children. The exposure-specific *HI* values are all well below 1, which implies that the elevated Hg concentrations in the examined media do not result in significant excess human health risks for all four considered scenarios. Moreover, the Σ*HI* values for the scenarios in both the city of Pavlodar and the CAP-affected zone indicate that the Hg contamination caused by the CAP does not pose an excess non-carcinogenic risk to the urban and rural population.

In the case of normal exposure, Σ*HI* values ranged from $6.20 \times 10^{-4}$ to $1.24 \times 10^{-2}$. The Σ*HI* values determined for conservative and maximum reasonable exposure cases (ranging from $9.30 \times 10^{-4}$ to $1.66 \times 10^{-2}$ and $1.73 \times 10^{-3}$ to $2.84 \times 10^{-2}$, respectively) did not substantially exceed these values. Thus, for all three exposure types, the total risk values were still in the orders of magnitude ranging from $10^{-4}$ to $10^{-2}$. Although the non-carcinogenic risk posed to urban and rural populations is considered acceptable, they still indicate that the residents of the Pavlodar region are directly exposed to Hg concentrations up to 2.84% of the *RfD*, which represents the estimated maximum daily intake that is unlikely to result in detrimental effects on human health, on a daily or weekly basis [17].

The determined risk values may be a concerning indicator as people in the vicinity of the CAP would not only be exposed to Hg emissions but also to other PTEs due to emissions from other plants in the industrial hub in Northern Kazakhstan [26]. The results indicated the presence of elevated concentrations of As, Ba, Cd, Cr, Co, Cu, Mn, Ni, Pb, Zn, and Se in soils and sediments at a small number of locations, i.e., exact hot spots of soil contamination [26]. Since the concentrations of these PTEs exceeded the Canadian and Kazakhstani background levels and maximum permissible concentrations (MPCs), further research on the origins, environmental impacts, and human health risk characterization for these non-biodegradable, persistent, and highly toxic pollutants is recommended [1,26]. In the context of potential carcinogenic and non-carcinogenic risks posed to the local population due to exposure to PTEs, such as highly toxic As, it is recommended to focus research efforts on the multi-element industrial contamination in the region [52]. Macro-evaluation of environmental and human health risks resulting from exposure to multiple

PTEs may present a more accurate understanding of the extent of ecosystem and human exposure to environmental contamination due to the operation of the industrial hub.

The calculated *HI* values could be used to determine the pathways that contribute most to human health risks from exposure to Hg in the environment (Table A8). For all scenarios and for both age groups, the inhalation pathway added most significantly to the risk, with a relative contribution from 11.9% to 100%. In the scenarios of recreational activities and fishing, the soil ingestion pathway also considerably contributed to risk (up to 92.0%), whereas its impact on the overall Hg exposure in the remaining two scenarios was low (<5.39%). The *HI* values determined for the soil ingestion pathway in the CAP-affected zone were different from the risk values determined in 1997–1998 and 2001–2002 using the CLEA and RISC-Human models, respectively [20]. The soil ingestion-specific risk values determined in the present study were lower than previously calculated *HI*, which ranged from 1.01 to 13.7. It should be noted that while on-site Hg concentrations were used by Woodruff and Dack [20] (e.g., average Hg soil concentration reported as 835.9 mg/kg), it was not possible to sample directly on-site (i.e., only the impacted zone around the CAP was sampled, resulting in an average soil concentration of 0.564 mg/kg) in the present study due to access prohibitions. Lower *HI* values in the present study may also be due to the decrease in the contamination level in the region. That being said, a direct comparison of these risk values may not be appropriate due to the differences in the exposure parameters and risk calculation algorithms. Finally, the dermal contact contributed to a very small extent (0.17–1.13%) in the scenarios where it was considered, which may be attributed to relatively low concentrations of Hg in Balkyldak Lake. Moreover, the low dermal bioaccessibility of Hg assumed in calculations may be the reason for this exposure pathway's low contribution to ΣHI. The surface water ingestion pathway contributed the least for almost all the scenarios, with a relative contribution to overall Hg exposure ranging from 0.01% to 0.55%.

The considerable contribution of the air inhalation pathway to the calculated ΣHI values is of concern due to the inevitability of the population's exposure to atmospheric Hg. The important role of the air inhalation pathway in the present study contrasts with the conventional dominance of the soil ingestion, which is usually accompanied by the inhalation pathway contributing the least to ΣHI values [1,11,53–56]. Some studies [42], however, have found that air inhalation may be a dominant contributor to human exposure to Hg in the case of adults, with ingestion still having more impact on ΣHI values for children, which is consistent with the results of the present study. Therefore, it is recommended to conduct further research on the atmospheric Hg concentration dynamics to determine the long-term effects of the population's exposure to Hg in air. It is important to note that the sum of the relative contributions is not always equal to 100, which is related to the fact that ΣHI values were determined using Monte-Carlo simulations and not through the summation of the individual *HI* values.

Interestingly, the ΣHI values for the urban residential scenario ($1.23 \times 10^{-2}$ to $2.54 \times 10^{-2}$) were greater than those for the rural population's exposure in a zone directly affected by the CAP emissions. It is, however, necessary to point out that the residential scenario indicates much longer exposure to Hg in soils and air, thereby increasing the ΣHI values. For the scenarios relevant to the CAP-affected zone, the *HI* and ΣHI values for the fishing scenario were close to the respective values for the scenario of residential exposure as they ranged from $8.10 \times 10^{-3}$ to $2.84 \times 10^{-2}$. In this scenario, longer and more frequent exposure to contaminated media is considered as the main reason for higher *HI* and ΣHI values.

The analysis of the extent of contributions of the parameters presented in the form of PDFs yielded results that differed from one age group to another and between the exposure scenarios. In the scenarios of the recreational activities, residential exposure, and fishing, for each of the exposure pathways, the Hg concentrations contributed the most to the resulting *HI* and ΣHI values. For the cases of exposure through ingestion of surface water, the second most contributing parameter was the contact rate. Despite not being the primary contributing parameter for the soil ingestion pathway in the fishing

and recreational scenarios, the ingestion rate also contributed to a great extent to the soil ingestion *HI* values. This considerable contribution of IR in the context of Monte Carlo simulations has been previously observed in studies on human exposure to heavy metals, including As, Pb, Hg, Cd, and Cr [11]. Therefore, the ingestion rate tends to be the dominant factor in human exposure to Hg in soils. Bodyweight tended to play a more prominent role in the resulting *HI* and Σ*HI* values for children in the recreational scenario, while the EF was contributing to all the *HI* values. Given that the sensitivity of human health risks for children to bodyweight values has been observed in several studies on human exposure to PTEs, it may be beneficial to further research the children's exposure to contaminated compartments, which may be of greater significance since the children have higher ingestion rates due to more frequent hand-to-mouth and soil pica behavior [11,14]. The EF and Hg concentrations in each of the compartments were the main contributors to the Σ*HI* values for children and adults in the recreational scenario. In the fishing scenario, however, the IR and Hg concentrations were the parameters to contribute the most to the Σ*HI* values, with EF primarily affecting the *HI* values for the air inhalation pathway.

In contrast to the fishing and recreational scenarios, the ingestion rate was the primary one for the soil ingestion pathway in the scenarios of the soil contact-intensive work, with the Hg concentrations in soils and air still having a great impact on the *HI* values. This change in the hierarchy order must be related to the higher level of soil contamination in the vicinity of the CAP. Similarly to the recreational scenario, the bodyweight was an important parameter for the *HI* and Σ*HI* values for children, while the EF parameter still came second in the parameter-impact hierarchy. EF was also the second most contributing parameter for the Σ*HI* values in the soil contact-intensive work scenario.

The determined pathway-specific *HI* and Σ*HI* values are also of importance despite being below 1 due to the possible transfer of Hg to other compartments. The extremely high concentrations of Hg in several surface water samples indicate that certain factors such as climatic conditions (e.g., windy weather) may cause the Hg amount in certain compartments to dramatically increase [21]. Given that previous studies on CAP-induced Hg contamination determined that the Hg concentrations were the highest in sediments and water in the vicinity of the plant's wastewater pipe, it is probable that the resuspension of sediment Hg and sediment–water interactions may result in the transfer of Hg from sediments to water, which may pose a greater risk due to further transfer of Hg to soils and air. An increase in aquatic, soil, and atmospheric Hg concentrations will increase the daily intake values of exposed populations, which may result in Σ*HI* values greater than 1. The most recent studies on the risk of Hg transfer between different environmental sinks in the CAP-affected zone were conducted more than a decade ago and were primarily dedicated to the assessment of the probability of Hg transfer to the groundwater systems in the vicinity of the CAP [21]. Thus, further research is needed to obtain more recent information on the Hg transport between air, water, soil, and sediments in the area.

It is also important to consider the Hg transfer between the environmental compartments and local biota. The results of the assessment of the local food chain contamination presented by Ullrich et al. [24] indicate that the Hg concentrations in fish in Balkyldak lake exceed the permissible levels and thus pose a risk to human health. Although it was prohibited to fish in Balkyldak lake to prevent human exposure to Hg-contaminated fish tissues, the enforcement measures seem not very effective. Therefore, the local population may still be exposed to elevated Hg concentrations through fish consumption. It is, therefore, important to conduct studies on the aquatic food chain contamination to determine an up-to-date extent of human exposure to Hg through the consumption of contaminated fish. Additionally, given that there are elevated Hg concentrations in soils and, subsequently, possibly in vegetation, it is recommended to study the extent of contamination of the terrestrial food chain to obtain a complete understanding of human oral exposure to Hg in the region. Its importance is magnified even more, given that the inhalation of Hg and the deposition of atmospheric Hg to vegetative cover may increase the concentration of Hg in animal and plant tissues that are consumed by the local population.

*3.4. Comparison to Literature*

A thorough evaluation of the literature on the Hg pollution episode in the Pavlodar region and the comparison of the current contamination levels with previously reported data have been presented in our previous studies [25,26]. To the best of our knowledge, no directly comparable studies aimed at providing comprehensive human health risk characterization exist, as no chemical assessment was completed on the former CAP, neither in scientific journal publications nor in technical reports. Nonetheless, several publications have addressed potential risks in different manners, and the main conclusions are summarized in Table A9. As previously discussed, the range for the calculated T*HI*s in the current study is lower than the values reported by Woodruff and Dack [20]. This is because on-site Hg concentrations were used in that study that are three orders of magnitude higher than the Hg soil concentrations reported in the present study, as it was not possible to sample directly on-site in the present study due to access prohibitions. Hg concentration values in fish tissue reported in the previous studies [24,57] also exceeded the permissible values, indicating potential human health risks for the local population consuming fish from Balkyldak Lake. Further investigation on the effect of fish consumption due to Hg methylation is recommended as previous studies are outdated and may not fully reflect the current situation. Moreover, it is worth mentioning that sampling directly on the territory of the former CAP was impossible due to access limitations; thus, we cannot confirm the absence of potential non-carcinogenic risks for workers of the Pavlodar industrial hub.

In addition to the evaluation of the literature on the Hg contamination in Pavlodar, the comparison to the studies investigating the impact of the relevant global Hg contamination cases has been previously reported [9,25,26]. However, there are few studies focusing on the evaluation of the potential human health risks resulting from the operations of the former CAPs. The values for Σ*HI*s in the current study are close to the values reported by Ferré-Huguet et al. [58] in Flix, Spain (Σ*HI* = 0.044 and 0.019 for children and adults, respectively). Operation of the former CAP in Spain caused major Hg pollution in riverbed material; however, the evaluation of the tap water did not show potential health risks for the local population (Σ*HI* = 0.048 and 0.020 for children and adults, respectively), despite reported high Hg concentration values in various media [58–60]. In contrast, high hazard values were reported in Augusta Harbor, Italy, where the former CAP operated in 1958–2003 [61]. High Hg concentrations detected in fish tissues (0.021–9.720 ng/g) indicate the bioaccumulation of Hg in the food chain, posing health risks for consumers [61]. Total hazard quotients were calculated based on USEPA *RfD* and WHO acceptable daily intake, with values ranging from 1.53–15.8 and 0.66–6.88, respectively, indicating the potential risks for the local population [61]. High *HI* values (0.02–2.35 for children and 0.02–1.46 for adults) were also reported by a study evaluating the impact of CAP in Botafogo, Brazil [62]. The main exposure pathway for both age categories was found to be the inhalation of Hg vapors, contributing from 80% to 92% to the non-carcinogenic risk value, which is similar to the results of the current study.

## 4. Conclusions

The present study focused on the exposure assessment and human health risk characterization for a population potentially exposed to elevated mercury (Hg) concentrations in the Pavlodar region, North Kazakhstan, in the vicinity of a former Hg-cell chlor-alkali plant (CAP). During the operational period from 1975 to 1993, discharges from the CAP peaked at approximately 1000 t of Hg, resulting in a major Hg contamination incident. The investigation covered the urban zone (Pavlodar city) as well as the area in the proximity of the former CAP, including nearby Balkyldak Lake, Irtysh River, and Sarymsak Pond. Human exposure to Hg was assessed by considering relevant pathways (soil and dust ingestion, surface water ingestion, air inhalation, surface water dermal contact) under four scenarios (recreational activities, fishing, residential exposure, soil contact-intensive work). The contributions of pathways to the total hazard index (Σ*HI*, denotes non-carcinogenic risk) ranked in the following order: air inhalation > soil and dust ingestion > surface

water dermal contact > surface water ingestion. According to risk calculations, the air inhalation route generated the most considerable risk (up to 100% of $\Sigma HI$). It is concerning that elemental Hg is still found in elevated concentrations and that the inhalation pathway contributes the most to risk, almost 30 years after the CAP has been closed. $\Sigma HI$ was acceptable for all four scenarios: recreational activities and fishing for the CAP-affected zone as well as residential exposure and soil contact-intensive work for the urban zone. Although Hg concentrations around the former CAP zone were generally higher than in the nearby impacted urban area, the total risk values for the residential exposure scenario were greater in comparison to the scenarios considered in the CAP-affected zone. The combined risk from all pathways was acceptable (deterministic assessment: $\leq 1.24 \times 10^{-2}$ for normal, $\leq 1.66 \times 10^{-2}$ for conservative, $\leq 2.84 \times 10^{-2}$ for maximum reasonable exposure, and $\leq 0.125$ for maximum concentration exposure) among all scenarios. The results for *HI* indicated that despite having elevated Hg levels in all compartments (soil, air, surface water, groundwater), the area surrounding the CAP was safe regarding Hg exposure for both urban and rural population subgroups (children and adults). That being said, the present study has two limitations. First, it must be noted that an assessment of contamination directly on the former CAP site on the industrial complex could not be performed due to access prohibitions; thus, the results cover only the region around the former CAP. Second, the present study did not consider the risks of ingesting contaminated fish, as fish were not characterized for their methyl-Hg content. Therefore, without an additional risk assessment, it is not possible to conclude the risks for the scenarios of Hg exposure among workers in the industrial complex (where the former CAP was also located) and of the local population consuming fish from the contaminated Lake Balkyldak. Finally, we recommend studies on the fate and transport of Hg in the contaminated ecosystem that will take Hg methylation and subsequent bioaccumulation in the food chain into account.

**Supplementary Materials:** The following are available online at https://www.mdpi.com/article/10.3390/su132413816/s1. Additional information on exposure assessment and risk characterization is available. In more detail: Table S1. Exposure parameters and their deterministic values for inhalation pathway used to assess exposure to Hg; Table S2. Exposure parameters and their deterministic values for soil ingestion pathway used to assess exposure to Hg; Table S3. Exposure parameters and their deterministic values for water ingestion pathway used to assess exposure to Hg; Table S4. Exposure parameters and their deterministic values for water dermal contact pathway used to assess exposure to Hg; Table S5. Daily intake values from deterministic exposure assessment of adults to Hg-contaminated soils for inhalation, dermal, and ingestion pathways under different exposure scenarios; Table S6. Daily intake values from deterministic exposure assessment of children to Hg-contaminated soils for inhalation, dermal, and ingestion pathways under different exposure scenarios; Table S7. Daily intake parameter values from probabilistic exposure assessment of adults to Hg-contaminated soils for inhalation, dermal, and ingestion pathways under different exposure scenarios; Table S8. Daily intake parameter values from probabilistic exposure assessment of children to Hg-contaminated soils for inhalation, dermal, and ingestion pathways under different exposure scenarios.

**Author Contributions:** Conceptualization, R.K., A.T., A.K., Z.A., F.K. and M.G.; Data curation, R.K., A.T., A.K. and Z.A.; Formal analysis, R.K., A.T., A.K., Z.A. and M.G.; Funding acquisition, F.K. and M.G.; Investigation, S.K., R.K., A.T., A.K., Z.A. and M.G.; Methodology, S.K., A.K., Z.A. and M.G.; Project administration, S.K., F.K. and M.G.; Resources, M.G.; Software, S.K., R.K. and M.G.; Supervision, S.K., A.T., F.K. and M.G.; Writing—original draft, S.K., R.K. and A.T.; Writing—review and editing, S.K., R.K., A.T., A.K., Z.A., F.K. and M.G. All authors have read and agreed to the published version of the manuscript.

**Funding:** The present research was supported by Nazarbayev University Faculty Development Competitive Research Grant Program (FDCRGP) (funder project reference: 090118FD5319). The article processing charge was funded by Nazarbayev University Social Policy Grant (SPG) Program.

**Institutional Review Board Statement:** Not applicable.

**Informed Consent Statement:** Not applicable.

**Data Availability Statement:** Data are contained within the article or Supplementary Material.

**Conflicts of Interest:** The authors declare that they have no known competing financial interests or personal relationships that could have appeared to influence the work reported in this paper.

## Appendix A

**Table A1.** Formulas and exposure parameters used to assess exposure to Hg.

| Exposure Scenarios | Exposure Pathway and Formula | Parameter |
|---|---|---|
| Recreational Activities, Fishing, Residential Exposure, Soil Contact-Intensive Work | Soil and Dust Ingestion:<br>$CDI_{sing} = (CS \times IR \times CF \times FI \times EF \times ED)/(BW \times AT)$ | $CDI_{sing}$ = CDI from Soil Ingestion (mg/kg·day)<br>CS = Soil Concentration (mg/kg)<br>IR = Ingestion Rate (mg soil/day)<br>CF = Conversion Factor (kg/mg)<br>FI = Fraction Ingested from the Contaminated Source (unitless)<br>EF = Exposure Frequency (days/year)<br>ED = Exposure Duration (years)<br>BW = Body Weight (kg)<br>AT = Averaging Time (days) |
| Residential Exposure, Fishing | Air Inhalation (chronic):<br>$EC_{ainh} = (CA \times ET \times EF \times ED)/AT$ | $EC_{ainh}$ = Exposure Concentration from Air Inhalation ($\mu g/m^3$)<br>CA = Contaminant Concentration in Air ($\mu g/m^3$)<br>ET = Exposure Time (hours/day)<br>EF = Exposure Frequency (days/year)<br>ED = Exposure Duration (years)<br>AT = Averaging Time (hours/exposure period) |
| Recreational Activities, Soil Contact-Intensive Work | Air Inhalation (sub-chronic):<br>$EC_{ainh} = (CA \times ET \times EF \times ED)/AT$ | $EC_{ainh}$ = Exposure Concentration from Air Inhalation ($\mu g/m^3$)<br>CA = Contaminant Concentration in Air ($\mu g/m^3$)<br>ET = Exposure Time (hours/day)<br>EF = Exposure Frequency (days/week)<br>ED = Exposure Duration (weeks)<br>AT = Averaging Time (hours) |
| Recreational Activities, Fishing | Surface Water Ingestion:<br>$CDI_{wing} = (CW \times CR \times ET \times EF \times ED)/(BW \times AT)$ | $CDI_{wing}$ = CDI from Water Ingestion (mg/kg·day)<br>CW = Chemical Concentration in water (mg/L)<br>CR = Contact Rate (L/h)<br>ET = Exposure Time (h/event)<br>EF = Exposure Frequency (events/yr)<br>ED = Exposure Duration (yr)<br>BW = body weight (kg)<br>AT = Averaging Time (period over which exposure is averaged) |
| Recreational Activities, Fishing | Surface Water Dermal Contact:<br>$DAD_{swder} = (DA_{event} \times EV \times ED \times EF \times SA)/(BW \times AT)$ | $DAD_{swder}$ = Dermally Absorbed Dose from Dermal Contact with Surface Water (mg/kg·day)<br>$DA_{event}$ = Absorbed Dose per Event ((mg/cm$^2$)/event)<br>$K_p$ = Dermal Permeability of a Compound in Water (cm/hour)<br>$C_w$ = Chemical Concentration in Water (mg/cm$^3$)<br>$t_{event}$ = Event Duration (hours/event)<br>EV = Event Frequency (events/day)<br>ED = Exposure Duration (years)<br>EF = Exposure Frequency (days/year)<br>SA = Skin Surface Area Available for Contact (cm$^2$)<br>BW = Body Weight (kg)<br>AT = Averaging Time (days) |

**Table A2.** Summary of concentrations of Hg (soils, sediments, water, air) with selected descriptive statistics.

| Medium | Average | Standard Deviation | Min | 25th Percentile | 50th Percentile | 75th Percentile | 95th Percentile | Max |
|---|---|---|---|---|---|---|---|---|
| Soils, sediments (mg/kg) | 0.564 | 2.34 | 0.0006 | 0.0092 | 0.0208 | 0.141 | 3.6805 | 24.1 |
| Surface water, groundwater (mg/L) | 0.093 | 0.259 | $4.00 \times 10^{-6}$ | $7.00 \times 10^{-6}$ | $1.70 \times 10^{-5}$ | 0.0003 | 0.5097 | 1.34 |
| Air (ng/m$^3$) | 7.46 | 6.25 | 1.00 | 3.00 | 5.50 | 9.0 | 22.0 | 37.0 |

**Table A3.** Percentiles (50th, 75th, 95th, and maximum concentration) of *HI* values from deterministic exposure assessment of adults to Hg-contaminated soils for inhalation, dermal, and ingestion pathways under different exposure scenarios.

| | | Percentile | | | | | | | | | | | | | | |
|---|---|---|---|---|---|---|---|---|---|---|---|---|---|---|---|---|
| | | Normal Exposure, 50th | | | | | Conservative Exposure, 75th | | | | | Maximum Reasonable Exposure, 95th | | | | |
| Scenario | | $HI_{sing}$ | $HI_{wing}$ | $HI_{inh}$ | $HI_{der}$ | $\Sigma HI$ | $HI_{sing}$ | $HI_{wing}$ | $HI_{inh}$ | $HI_{der}$ | $\Sigma HI$ | $HI_{sing}$ | $HI_{wing}$ | $HI_{inh}$ | $HI_{der}$ | $\Sigma HI$ |
| CAP-affected zone | Recreational Activities | $1.49 \times 10^{-5}$ | $3.61 \times 10^{-6}$ | $8.93 \times 10^{-4}$ | $1.83 \times 10^{-5}$ | $9.30 \times 10^{-4}$ | $1.17 \times 10^{-4}$ | $1.77 \times 10^{-5}$ | $1.49 \times 10^{-3}$ | $4.45 \times 10^{-5}$ | $1.67 \times 10^{-3}$ | $8.60 \times 10^{-4}$ | $1.63 \times 10^{-4}$ | $3.00 \times 10^{-3}$ | $1.14 \times 10^{-4}$ | $4.13 \times 10^{-3}$ |
| | Fishing | $1.79 \times 10^{-4}$ | $2.56 \times 10^{-5}$ | $4.93 \times 10^{-3}$ | $6.37 \times 10^{-5}$ | $5.20 \times 10^{-3}$ | $9.09 \times 10^{-3}$ | $1.75 \times 10^{-4}$ | $8.22 \times 10^{-3}$ | $1.57 \times 10^{-4}$ | $1.76 \times 10^{-2}$ | $2.83 \times 10^{-2}$ | $3.95 \times 10^{-4}$ | $1.65 \times 10^{-2}$ | $3.99 \times 10^{-4}$ | $4.57 \times 10^{-2}$ |
| Urban zone | Residential Exposure | $7.28 \times 10^{-6}$ | | $1.28 \times 10^{-2}$ | | $1.28 \times 10^{-2}$ | $3.66 \times 10^{-5}$ | | $1.60 \times 10^{-2}$ | | $1.60 \times 10^{-2}$ | $1.12 \times 10^{-3}$ | | $2.80 \times 10^{-2}$ | | $2.91 \times 10^{-2}$ |
| | Soil Contact-Intensive Work | $5.26 \times 10^{-5}$ | | $3.17 \times 10^{-3}$ | | $3.23 \times 10^{-3}$ | $2.65 \times 10^{-4}$ | | $3.97 \times 10^{-3}$ | | $4.23 \times 10^{-3}$ | $1.04 \times 10^{-3}$ | | $6.94 \times 10^{-3}$ | | $7.99 \times 10^{-3}$ |

**Table A4.** Percentiles (50th, 75th, and 95th) of *HI* values from deterministic exposure assessment of children to Hg-contaminated soils for inhalation, dermal, and ingestion pathways under different exposure scenarios.

| | | Percentile | | | | | | | | | | | | | | |
|---|---|---|---|---|---|---|---|---|---|---|---|---|---|---|---|---|
| | | Normal Exposure, 50th | | | | | Conservative Exposure, 75th | | | | | Maximum Reasonable Exposure, 95th | | | | |
| Scenario | | $HI_{sing}$ | $HI_{wing}$ | $HI_{inh}$ | $HI_{der}$ | $\Sigma HI$ | $HI_{sing}$ | $HI_{wing}$ | $HI_{inh}$ | $HI_{der}$ | $\Sigma HI$ | $HI_{sing}$ | $HI_{wing}$ | $HI_{inh}$ | $HI_{der}$ | $\Sigma HI$ |
| CAP-affected zone | Recreational Activities | $2.44 \times 10^{-4}$ | $3.09 \times 10^{-5}$ | $8.93 \times 10^{-4}$ | $3.09 \times 10^{-5}$ | $1.20 \times 10^{-3}$ | $1.98 \times 10^{-3}$ | $1.52 \times 10^{-4}$ | $1.49 \times 10^{-3}$ | $7.74 \times 10^{-5}$ | $3.70 \times 10^{-3}$ | $1.39 \times 10^{-2}$ | $6.00 \times 10^{-4}$ | $3.00 \times 10^{-3}$ | $1.99 \times 10^{-4}$ | $1.77 \times 10^{-2}$ |
| Urban zone | Residential Exposure | $6.48 \times 10^{-5}$ | | $1.28 \times 10^{-2}$ | | $1.28 \times 10^{-2}$ | $3.44 \times 10^{-4}$ | | $1.60 \times 10^{-2}$ | | $1.63 \times 10^{-2}$ | $1.01 \times 10^{-2}$ | | $2.80 \times 10^{-2}$ | | $3.81 \times 10^{-2}$ |

**Table A5.** Relative contributions of different exposure pathways to *HI* values for children and adults in deterministic risk assessment.

| | HI (Relative Contribution) | | | | | | | | | | | | | | | | | |
|---|---|---|---|---|---|---|---|---|---|---|---|---|---|---|---|---|---|---|
| | Recreational Activities | | | | | | Fishing | | | Residential Exposure | | | | | | Soil Contact-Intensive Work | | |
| Pathway | $HI_{50\%}$ | | $HI_{75\%}$ | | $HI_{95\%}$ | | $HI_{50\%}$ | $HI_{75\%}$ | $HI_{95\%}$ | $HI_{50\%}$ | | $HI_{75\%}$ | | $HI_{95\%}$ | | $HI_{50\%}$ | $HI_{75\%}$ | $HI_{95\%}$ |
| | Children | Adults | Children | Adults | Children | Adults | | Adults | | Children | Adults | Children | Adults | Children | Adults | | Adults | |
| Air Inhalation | 74.4% | 96.0% | 40.3% | 89.2% | 17.0% | 72.6% | 94.8% | 46.6% | 36.2% | 99.5% | 99.9% | 98.2% | 99.8% | 73.5% | 96.2% | 98.4% | 93.8% | 87.0% |
| Soil Ingestion | 20.3% | 1.60% | 53.5% | 7.01% | 78.5% | 20.8% | 3.5% | 51.5% | 62.0% | 0.51% | 0.06% | 2.11% | 0.23% | 26.5% | 3.85% | 1.63% | 6.25% | 13.0% |
| Water Ingestion | 2.58% | 0.39% | 4.11% | 1.06% | 3.39% | 3.95% | 0.49% | 0.99% | 0.86% | | | | | | | | | |
| Water Dermal Contact | 2.58% | 1.97% | 2.09% | 2.66% | 1.12% | 2.76% | 1.23% | 0.89% | 0.87% | | | | | | | | | |

**Table A6.** Percentiles (50th, 75th, and 95th) of *HI* values from probabilistic exposure assessment of adults to Hg-contaminated soils for inhalation, dermal, and ingestion pathways under different exposure scenarios.

| Scenario | | Percentile | | | | | | | | | | | | | | |
|---|---|---|---|---|---|---|---|---|---|---|---|---|---|---|---|---|
| | | Normal Exposure, 50th | | | | | Conservative Exposure, 75th | | | | | Maximum Reasonable Exposure, 95th | | | | |
| | | $HI_{sing}$ | $HI_{wing}$ | $HI_{inh}$ | $HI_{der}$ | $\Sigma HI$ | $HI_{sing}$ | $HI_{wing}$ | $HI_{inh}$ | $HI_{der}$ | $\Sigma HI$ | $HI_{sing}$ | $HI_{wing}$ | $HI_{inh}$ | $HI_{der}$ | $\Sigma HI$ |
| CAP-Affected zone | Recreational Activities | $6.25 \times 10^{-5}$ | $7.06 \times 10^{-8}$ | $4.86 \times 10^{-4}$ | $2.00 \times 10^{-6}$ | $6.20 \times 10^{-4}$ | $1.64 \times 10^{-4}$ | $1.80 \times 10^{-7}$ | $7.39 \times 10^{-4}$ | $4.08 \times 10^{-6}$ | $9.30 \times 10^{-4}$ | $6.36 \times 10^{-4}$ | $6.87 \times 10^{-7}$ | $1.33 \times 10^{-3}$ | $1.10 \times 10^{-5}$ | $1.73 \times 10^{-3}$ |
| | Fishing | $1.05 \times 10^{-3}$ | $1.98 \times 10^{-5}$ | $5.65 \times 10^{-3}$ | $7.69 \times 10^{-5}$ | $8.10 \times 10^{-3}$ | $3.42 \times 10^{-3}$ | $4.92 \times 10^{-5}$ | $8.37 \times 10^{-3}$ | $1.47 \times 10^{-4}$ | $1.26 \times 10^{-2}$ | $1.91 \times 10^{-2}$ | $1.90 \times 10^{-2}$ | $1.47 \times 10^{-4}$ | $3.87 \times 10^{-4}$ | $2.84 \times 10^{-2}$ |
| Urban Zone | Residential Exposure | $2.81 \times 10^{-6}$ | | $1.23 \times 10^{-2}$ | | $1.23 \times 10^{-2}$ | $7.58 \times 10^{-6}$ | | $1.65 \times 10^{-2}$ | | $1.65 \times 10^{-2}$ | $3.07 \times 10^{-5}$ | | $2.53 \times 10^{-2}$ | | $2.52 \times 10^{-2}$ |
| | Soil Contact-Intensive Work | $1.89 \times 10^{-5}$ | | $3.05 \times 10^{-3}$ | | $3.15 \times 10^{-3}$ | $6.16 \times 10^{-5}$ | | $4.14 \times 10^{-3}$ | | $4.30 \times 10^{-3}$ | $3.76 \times 10^{-4}$ | | $6.60 \times 10^{-3}$ | | $6.79 \times 10^{-3}$ |

**Table A7.** Percentiles (50th, 75th, and 95th) of *HI* values from probabilistic exposure assessment of children to Hg-contaminated soils for inhalation, dermal, and ingestion pathways under different exposure scenarios.

| Scenario | | Percentile | | | | | | | | | | | | | | |
|---|---|---|---|---|---|---|---|---|---|---|---|---|---|---|---|---|
| | | Normal Exposure, 50th | | | | | Conservative Exposure, 75th | | | | | Maximum Reasonable Exposure, 95th | | | | |
| | | $HI_{sing}$ | $HI_{wing}$ | $HI_{inh}$ | $HI_{der}$ | $\Sigma HI$ | $HI_{sing}$ | $HI_{wing}$ | $HI_{inh}$ | $HI_{der}$ | $\Sigma HI$ | $HI_{sing}$ | $HI_{wing}$ | $HI_{inh}$ | $HI_{der}$ | $\Sigma HI$ |
| CAP-Affected Zone | Recreational Activities | $9.95 \times 10^{-4}$ | $3.15 \times 10^{-7}$ | $4.84 \times 10^{-4}$ | $3.38 \times 10^{-6}$ | $1.66 \times 10^{-3}$ | $2.58 \times 10^{-3}$ | $7.98 \times 10^{-7}$ | $7.39 \times 10^{-4}$ | $6.89 \times 10^{-6}$ | $3.22 \times 10^{-3}$ | $1.03 \times 10^{-2}$ | $3.02 \times 10^{-6}$ | $1.33 \times 10^{-3}$ | $1.85 \times 10^{-5}$ | $1.12 \times 10^{-2}$ |
| Urban Zone | Residential Exposure | $4.35 \times 10^{-5}$ | | $1.23 \times 10^{-2}$ | | $1.24 \times 10^{-2}$ | $1.16 \times 10^{-4}$ | | $1.65 \times 10^{-2}$ | | $1.66 \times 10^{-2}$ | $4.77 \times 10^{-4}$ | | $2.53 \times 10^{-2}$ | | $2.54 \times 10^{-2}$ |

**Table A8.** Relative contributions of different exposure pathways to *HI* values for children and adults in probabilistic risk assessment.

| Pathway | Relative Contribution | | | | | | | | | | | | | | | | | | |
|---|---|---|---|---|---|---|---|---|---|---|---|---|---|---|---|---|---|---|---|
| | Recreational Activities | | | | | | Fishing | | | Residential Exposure | | | | | | Soil Contact-Intensive Work | | |
| | $HI_{50\%}$ | | $HI_{75\%}$ | | $HI_{95\%}$ | | $HI_{50\%}$ | $HI_{75\%}$ | $HI_{95\%}$ | $HI_{50\%}$ | | $HI_{75\%}$ | | $HI_{95\%}$ | | $HI_{50\%}$ | $HI_{75\%}$ | $HI_{95\%}$ |
| | Children | Adults | Children | Adults | Children | Adults | Adults | Adults | Adults | Children | Adults | Children | Adults | Children | Adults | Adults | Adults | Adults |
| Air Inhalation | 29.2% | 78.4% | 23.0% | 79.5% | 11.9% | 76.9% | 69.8% | 66.4% | 42.8% | 99.2% | 100% | 99.4% | 100% | 99.6% | 99.9% | 96.8% | 96.5% | 94.6% |
| Soil Ingestion | 59.9% | 10.1% | 80.1% | 17.6% | 92.0% | 36.8% | 13.0% | 27.1% | 55.6% | 0.35% | 0.02% | 0.70% | 0.05% | 1.88% | 0.12% | 0.60% | 1.44% | 5.39% |
| Water Ingestion | 0.02% | 0.01% | 0.02% | 0.02% | 0.03% | 0.04% | 0.24% | 0.39% | 0.55% | | | | | | | | | |
| Water Dermal Contact | 0.20% | 0.32% | 0.21% | 0.44% | 0.17% | 0.64% | 0.95% | 1.17% | 1.13% | | | | | | | | | |

**Table A9.** Literature on impact of Hg contamination in the Pavlodar region.

| Study | Analysis Method and Results | Main Remarks |
|---|---|---|
| Mercury (Hg) contamination of fish fauna of Balkyldak technical pond [57] | Hg in fish tissues:<br>– Dace: 4.36 mg/kg<br>– Crucian carp: 3.18 mg/kg<br>– Tench: 1.98 mg/kg | Hg concentrations in 50 out of 55 samples exceeded the maximum allowable concentration of 0.3 mg/kg. |
| Analysis of risk from mercury contamination at the Khimprom Plant in Kazakhstan [20] | – Soil: 0.0067–835.9 mg/kg<br>– Groundwater: 0.00022–18 mg/L<br>– HQ: 1.01–6.67 (CLEA model), 1.93–5080 (RISC-Human model) | The hazard indices calculated by the CLEA and RISC-Human models indicate the potential risks for human health. |
| Mercury contamination in the vicinity of a derelict chlor-alkali plant. Part II: Contamination of the aquatic and terrestrial food chain and potential risks to the local population [24] | – Soil: 0.10–3.30 mg/kg<br>– Groundwater: <5 ng/L<br>– Bovine tissues: 10.96 µg/kg<br>– Bovine milk: <2 µg/kg<br>– Fish tissues: 0.16–2.2 mg/kg | 91% of fish tissue concentrations exceed the permissible level indicating the dietary Hg intake is the primary exposure route to mercury, posing potential human health risks. |
| Heavy metals accumulation in children hair [23] | – Hair samples of 12–14-year-old children: 0.20–0.70 mg/kg | The highest Hg concentration values were observed in parts of the city located closer to the CAP. |
| The role of qualitative risk assessment in environmental management: A Kazakhstani case study [63] | Qualitative risk assessment via interviewing:<br>– Scientists: atmospheric Hg concentrations are of concern and have to be consistently monitored<br>– Stakeholders: Hg contamination is not of concern anymore<br>– Local population: groundwater, soil, and fish tissue contamination are of primary concern | Stakeholder and local population perceptions on human health risks associated with Hg contamination are the major obstacles for further remediation actions in the region, which may result in adverse health effects for the site workers and local people. |
| The present study | Calculated $\Sigma HI$ values ranged:<br>– from $1.20 \times 10^{-3}$ to $2.32 \times 10^{-2}$ for children<br>– from $6.20 \times 10^{-4}$ to $2.72 \times 10^{-2}$ for adults | The determined $\Sigma HI$ values indicate acceptable human health risks from exposure to mercury contamination in the region. Air inhalation and soil ingestion pathways are the main Hg exposure routes in the analyzed four scenarios. |

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
