# Peer review of "Potential Human Exposure to Mercury (Hg) in a Chlor-Alkali Plant Impacted Zone: Risk Characterization Using Updated Site Assessment Data"

_sustainability, doi:10.3390/su132413816_

Round 1
Reviewer 1 Report
The present manuscript provided Hg health risk assessment within 4 scenarios, including recreational activities, fishing, redidential exposure, and soil contact-intensive work based on published data in Povlodar, Kazakhstan. Overall, the present study was written in clear English and understandable logic. However, if all data in the present manuscript was cited from the previously publihsed paper, it is no need to repeat the sampling and determination in the Materials and Methods section. Meanwhile, if so, the present work is like re-analyze on the published data, which reduce the innovation here. Furthermore, the implications of present work is quite local since the study site is not very typical and the health risk is very low now.
Since no line number was provided in the present version, the specific comments were marked in the manuscript file, which was attached here. Please check. Thanks.

Reviewer 2 Report
The manuscript entitled “Potential human exposure to mercury (Hg) in a chlor-al-kali plant impacted zone: Risk characterization using updated site assessment data” is well written, and the authors have selected the literature well and have chosen a contemporary time range that makes the manuscript relevant. I have no hesitation in accepting this manuscript for publication.
Author Response
We thank the Reviewer for their time and effort, and for accepting our manuscript.
Reviewer 3 Report
The article is well documented. The results are clearly presented. The conclusions are transparent. The authors are aware of some inevitable shortcomings. Overall, I rate the article high. Please improve readability of tables 5,6,7
In my opinion, the article needs revision mainly of the tables as well as some language corrections. The text is well prepared and the indices are calculated correctly. I appreciate that the authors also described the difficulties and are aware of the research that should be refined in the future to fully describe the problem of mercury poisoning. in my opinion, the work is fine at the moment.
Round 2
Reviewer 1 Report
My main concerns were addressed. Thanks for the improvement on the manuscript.